# WeakSAM: Segment Anything Meets Weakly-supervised Instance-level Recognition

## ABSTRACT

Weakly-supervised visual recognition using inexact supervision is a critical yet challenging learning problem. It significantly reduces human labeling costs and traditionally relies on multi-instance learning and pseudo-labeling. This paper introduces WeakSAM and solves the weakly-supervised object detection (WSOD) and segmentation by utilizing the pre-learned world knowledge contained in a vision foundation model, i.e., the Segment Anything Model (SAM). WeakSAM addresses two critical limitations in traditional WSOD retraining, i.e., pseudo ground truth (PGT) incompleteness and noisy PGT instances, through adaptive PGT generation and Region of Interest (RoI) drop regularization. It also addresses the SAM's shortcomings of requiring human prompts and category unawareness in object detection and segmentation. Our results indicate that WeakSAM significantly surpasses previous state-of-the-art methods in WSOD and WSIS benchmarks with large margins, i.e. average improvements of 7.4% and 8.5%, respectively.

## CCS CONCEPTS

• Computing methodologies → Object detection; Image segmentation.

## KEYWORDS

Weakly-supervised Learning, Segment Anything Model, Object Detection, Instance Segmentation

## 1 INTRODUCTION

Weakly-supervised learning (WSL) [73, 74, 91] is a crucial component of machine learning. It is particularly valuable in tasks where strong supervision is difficult to annotate due to the high cost of data labeling [16, 47, 54]. Due to the massive demand for annotated data in visual perception, WSL is essential in developing a label-efficient recognition system. In the standard weakly-supervised visual perception paradigm [5, 8, 51, 56, 60, 63, 64, 75–77, 85, 87], training commences with inexact supervision, such as image-level labels. Subsequently, the trained WSL network is employed to generate pseudo ground truth (PGT), which serves as a form of refined, albeit still inaccurate supervision. Finally, the PGT is used as inaccurate supervision to launch WSL retraining. Although the iterative WSL process achieves significant progress, it is still limited by the

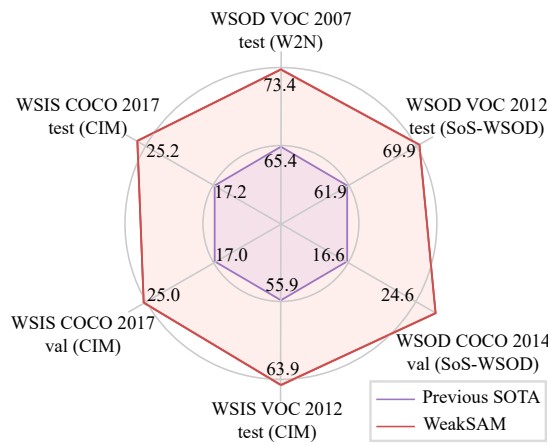

**Figure 1: Quantitative comparisons between WeakSAM and previous SOTA methods under different tasks and benchmarks. The scale of each axis in the radar chart is normalized by the performance of the previous SOTA methods (marked in parentheses), and the stride of each axis is the same.**

lack of external knowledge, which restricts the performance of WSL and hinders it from matching fully-supervised learning (FSL).

Nowadays, foundation models are gaining increasing attention because of their transferable pre-learned world knowledge, which can be regarded as powerful external knowledge for WSL. As a vision foundation model, SAM [34] achieves outstanding performance in interactive, class-agnostic segmentation. SAM owes its success to promptable training on a large-scale dataset. However, there are two main drawbacks to SAM: First, SAM requires interactive operations as input, which means it cannot work automatically without human prompts. Second, SAM produces class-agnostic segments and cannot assign class labels. These drawbacks severely restrict the application of SAM as a direct and generic visual framework. As a strong complement, WSL is good at mining classification clues through inexact supervision, which can provide automatic prompts for SAM. Subsequently, WSL with SAM's knowledge can further bring class-aware perception.

This motivates us to assimilate SAM within the WSL paradigm. The WeakSAM framework is designed to harness transferable knowledge from SAM, thereby enriching the WSL process. Simultaneously, it offers the capability to deliver automatic classification clues to SAM. This bidirectional enhancement constructs a promising foundation-model-based weakly-supervised visual perception framework. Specifically, in a weakly-supervised object detection (WSOD) setting, WeakSAM uses classification clues as SAM prompts to produce proposals automatically. These proposals are then used in WSOD training for class-aware perception.

Within the scope of the WeakSAM framework, our analysis identifies two prevailing limitations in the iterative WSOD retraining

approach: the issue of pseudo ground truth (PGT) incompleteness and the presence of noisy PGT instances. The former, PGT incompleteness, refers to the tendency of WSOD-generated PGT to omit some objects or categories, leading to insufficient training for these categories. The latter, noisy PGT instances, pertain to the prevalent presence of noise within the PGT, which adversely impacts the retraining process. To effectively mitigate these challenges, we introduce two key strategies: adaptive PGT generation to address the PGT incompleteness problem, and Region of Interest (RoI) drop regularization to counteract the noise in PGT instances. Moreover, WeakSAM's capability enables the extension in the realm of weakly-supervised instance segmentation (WSIS). In this context, SAM is employed to further refine WeakSAM-PGT, enabling the generation of pseudo instance segmentation labels. This approach exemplifies WeakSAM is promising to build a unified weakly-supervised instance-level recognition framework.

The main contributions of this paper can be summarized as follows:

- We propose a weakly-supervised instance-level recognition framework (WeakSAM), which automatically prompts SAM by classification clues for proposals. The WeakSAM proposals reduce the generation time by 65.5% and improve the recall (IoU=0.9) by 22.9%, compared to Selective Search [69].
- We analyze the weaknesses in traditional WSOD retraining, and propose adaptive PGT generation and RoI drop regularization to address them, respectively. After the WeakSAM-WSOD is complete, the proposed WeakSAM can be easily applied to WSIS further.
- The proposed WeakSAM achieves state-of-the-art (SOTA) results on the WSOD and WSIS benchmarks, significantly surpassing previous SOTA methods as shown in Fig. 1.

## 2 RELATED WORK

### 2.1 Segment Anything Model

The recent Segment Anything Model (SAM) [34] draws great attention from researchers. The SAM is trained on SA-1B with over 1 billion masks, following the model-in-the-loop manner. Besides, SAM performs superior zero-shot transfer capabilities and is applied in many visual tasks, e.g., FGVP [78] incorporates SAM to achieve zero-shot fine-grained visual prompting, MedSAM [48] adapts SAM into a large scale medical dataset to build a medical foundation model, and some methods [7, 30, 62] utilize SAM to deal with the weakly-supervised semantic segmentation problem. However, SAM is an interactive segmentation method, which heavily relies on human prompts.

In our approach, we innovatively propose to automatically prompt SAM using classification clues for extracting region proposals. This method results in high-recall proposals that surpass traditional methods like Selective Search in terms of both efficiency and effectiveness. This advancement represents a significant improvement in the domain of proposal generation within the WSOD framework.

### 2.2 Weakly-supervised Object Detection

Weakly-supervised object detection (WSOD) with image-level labels [2, 3, 12, 17, 29, 35, 40, 45, 61, 66, 70, 71, 86] is important for reducing the human annotation burden. The previous works, i.e.,

WSDDN [4] and OICR [65], proposed the Multiple Instance Learning and online refinement paradigms. The later works aimed to improve the WSOD performance from different perspectives. Such as WSOD$^2$ [81] introduced bottom-up object evidence, PCL [64] proposed to cluster proposals, MIST [53] utilized a self-training algorithm, etc. Besides, some methods [26, 31, 38, 60, 64, 88] also retrained a fully-supervised object detection network with generated pseudo ground truth (PGT). However, most of them used the proposals generated from low-level methods, i.e., Selective Search [69], EdgeBox [95], and MCG [50], which contain a great number of redundant proposals and bring an optimization challenge.

Different from previous methods, our WeakSAM proposals have fewer numbers and higher recall, which reduces the difficulty of finding the correct proposals for WSOD methods. For the key problem of PGT incompleteness and noisy PGT instances, we propose adaptive PGT generation and Region of Interest (RoI) drop regularization to address them, respectively.

### 2.3 Weakly-supervised Instance Segmentation

Weakly-supervised instance segmentation (WSIS) aims to achieve instance segmentation through weak supervision, such as box-level supervision [11, 24, 32, 37, 39, 42, 67, 72, 83, 93], and image-level supervision [19, 23, 25, 28, 36, 46, 49, 84, 94]. The WSIS with image-level supervision is challenging because it lacks accurate instance locations. Some image-level WSIS methods use class activation map (CAM) [89] to extract coarse object locations, such as PRM [90], IAM [94], IRNet [1], BESTIE [33], etc. Some other image-level WSIS methods try to incorporate instance clues from extra priors, such as Fan et al. [15], LIID [46], CIM [41], etc. However, they always need complicated networks and lack high-quality instance segments.

Different from previous WSIS methods, the proposed WSIS extension using WeakSAM PGT and SAM's prediction is concise and effective. The generated pseudo instance labels can further be applied to any fully-supervised instance segmentation method.

## 3 METHODS

We present the WeakSAM framework as shown in Fig. 2. At first, WeakSAM collects classification activations from a classification ViT. Subsequently, WeakSAM automatically generates prompts from classification activations and spatial samples. Next, WeakSAM sends the prompts to SAM for WeakSAM proposals. Then, we launch the weakly-supervised object detection (WSOD) pipeline, which is enhanced by WeakSAM proposals, adaptive pseudo ground truth (PGT) generation, and RoI drop regularization. Last, we use the SAM-enhanced pseudo instance labels to launch the weakly-supervised instance segmentation extension.

### 3.1 Classification Clues as Automatic Prompts

Previous WSOD methods face an optimization problem caused by the redundant proposals, e.g., Selective Search [69] and Edge-Box [95], because these proposals are only based on low-level features. To address this problem, we propose to transfer knowledge in the foundation model, i.e., SAM, for proposal generation. We use classification clues to prompt SAM automatically, which also solves the shortcoming of SAM requiring interactive prompts

*Classification Activation Generation.* As shown in Fig. 2, we extract classification clues from a classification ViT. Specifically, we

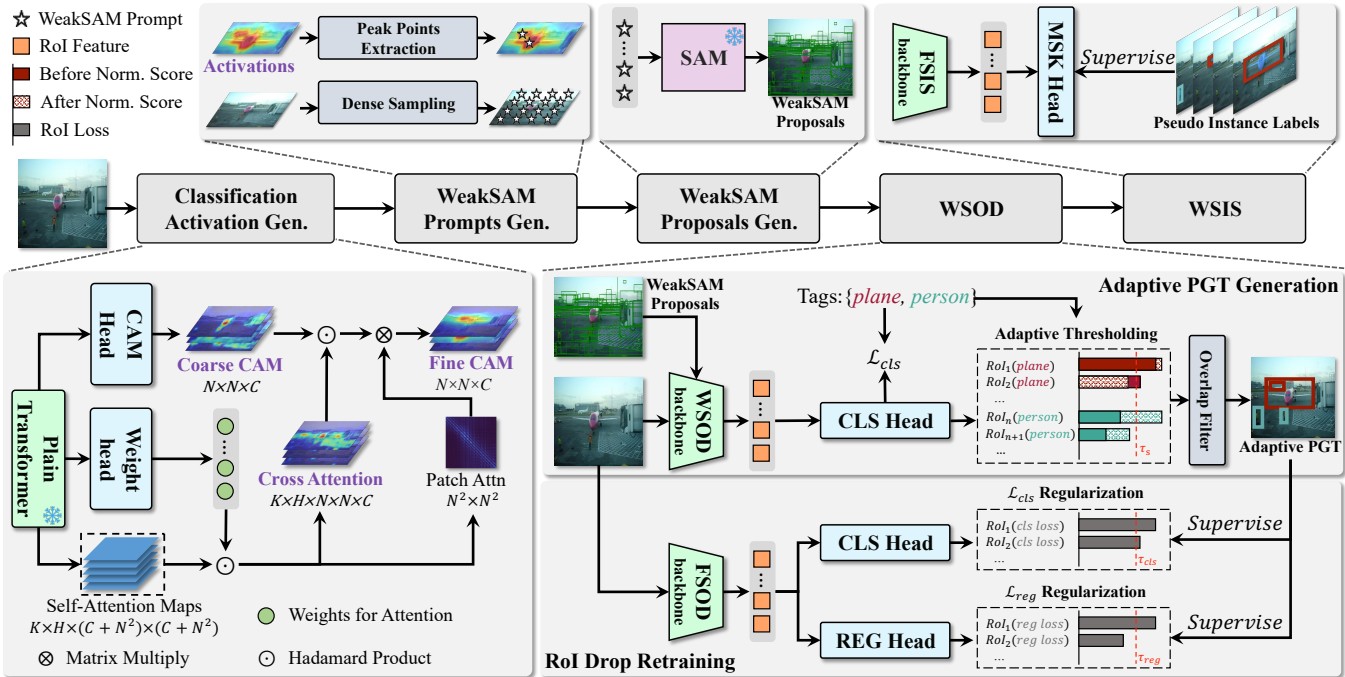

**Figure 2: An overview of the proposed WeakSAM framework. We first generate activation maps from a classification ViT [92]. Subsequently, we introduce classification clues and spatial points as automatic WeakSAM prompts, which address the problem of SAM requiring interactive prompts. Next, we use the WeakSAM proposals in the WSOD pipeline, in which the weakly-supervised detector performs class-aware perception to annotate pseudo ground truth (PGT). Then, we analyze the incompleteness and the noise problem existing in PGT and propose adaptive PGT generation, RoI drop regularization to address them, respectively. Finally, we launch WSIS training supervised by pseudo instance labels, which requires adaptive PGT as SAM prompts. The snowflake mark means the model is frozen.**

choose the pre-trained weakly-supervised semantic segmentation network, WeakTr [92], to provide classification clues because of its superior localization ability. At first, we extract cross-attention maps $\mathbf{CA} \in \mathbb{R}^{K \times H \times N \times N \times C}$ from the self-attention maps, where $K$ is the number of transformer encoding layers, $H$ is the number of attention heads in each layer, $N \times N$ is the spatial size of the visual tokens, and $C$ represents the total number of classification categories. Then, we obtain coarse class activation map (CAM) [89], $\mathbf{CAM}_{coarse} \in \mathbb{R}^{N \times N \times C}$, from the convolutional CAM head, which takes visual tokens at the final transformer layer as input and produces coarse CAM. Last, we use coarse CAM and weighted self-attention maps to produce fine CAM, $\mathbf{CAM}_{fine} \in \mathbb{R}^{N \times N \times C}$.

*WeakSAM Prompts Generation.* As shown in Fig. 2, we extract prompts from dense sampling points and activations, which include cross-attention maps, coarse CAM, and fine CAM. At first, the dense sampling requires splitting the image into $S \times S$ patches and taking the center points as prompts. Notably, the dense sampling points provide spatial-aware prompts but lack explicit reference to objects and semantics. Then, we get peak points from the cross-attention maps as prompts. We observe that these maps do not solely concentrate on objects from their corresponding categories but also give attention to objects from different categories. So, we mark these prompts as instance-aware ones. Last, we extract peak points from coarse CAM and fine CAM as semantic-aware prompts, which are more precise and focus on areas of foreground objects.

Specifically, we extract peak points from cross-attention maps and CAMs, as shown in Algorithm 1. Given cross-attention maps or CAMs as input, we first initialize the peak points list $P$, peak values list $V$, deleted lists $P_{\text{delete}}$, $V_{\text{delete}}$, and max pooling operation. Next, we reshape the input maps and ensure the last two dimensions correspond to the original image size and the others as the first dimension. Then, we apply max pooling on the input maps $M$, and sort $V$ and $P$ in descending order based on $V$. Last, we remove points with low activation values or close to high-score points.

*WeakSAM Proposals Generation.* At the WeakSAM proposal generation stage, we use the three kinds of prompts to prompt SAM automatically. We directly add semantic-aware prompts and spatial-ware prompts to the prompt list, because they usually have clear localization to foreground objects and spatial positions, respectively. For the instance-aware prompts that have some redundancy, we cluster them to filter the duplicated ones and then add them to the prompt list. Finally, the prompt list is used to prompt SAM for WeakSAM proposals.

## 3.2 WeakSAM WSOD Pipeline

To better describe the proposed weakly-supervised object detection (WSOD) pipeline, we first present the weakly-supervised detector training with WeakSAM proposals. Then, we identify the PGT incompleteness problem and introduce the proposed adaptive PGT generation to address it. Last, we analyze the noise problem existing

**Algorithm 1** Peak Points Extraction

**Require:** maps $\mathbf{M}$ (CA or CAM), kernel size $k$, activation threshold $\tau$
**Ensure:** peak points coordinates list $P = [p_0, p_1, \ldots, p_{n-1}]$, corresponding peak values list $V = [v_0, v_1, \ldots, v_{n-1}]$
  1: $\mathbf{M} = \mathbf{M}.\text{view}(-1, N, N)$ // reshape
  2: Initialize $P, V$ as empty list
  3: Initialize Maxpool() operation with kernel size $k$
  4: $P, V = \text{Maxpool}(\mathbf{M})$ // get coordinates and values
  5: Sort $V$ in descending order of numerical value, and rearrange $P$ accordingly
  6: Initialize list $P_{\text{delete}}, V_{\text{delete}}$ to mark points for deletion
  7: **for** each index $i$ from 0 to length($P$) **do**
  8:    // skip further checks for points marked for deletion
  9:    **if** $p_i$ in $P_{\text{delete}}$ **then**
10:       Continue
11:    **end if**
12:    // mark activation points with low score
13:    **if** $v_i < \tau$ **then**
14:       Append $p_i, v_i$ to $P_{\text{delete}}, V_{\text{delete}}$
15:       Continue
16:    **end if**
17:    // mark lower-score points near the current point
18:    **for** each index $j$ from $i + 1$ to length($P$) **do**
19:       **if** $||p_j - p_i|| \le k/2$ **then**
20:          Append $p_j, v_j$ to $P_{\text{delete}}, V_{\text{delete}}$
21:       **end if**
22:    **end for**
23: **end for**
24: Remove all points in $P_{\text{delete}}$ and $V_{\text{delete}}$ from $P$ and $V$
25: **return** $P, V$

in the retraining phase, and propose Region of Interest (RoI) drop regularization to alleviate the effect of noise.

*Weakly-supervised Detector Training.* A primary challenge in traditional WSOD methods is the low training efficiency, largely attributed to the redundancy of proposals. Traditional approaches often involve the Region of Interest pooling layer processing thousands of proposals per image, which impairs both effectiveness and efficiency. To address this issue, our WeakSAM proposals adopt transferred knowledge from SAM and classification clues. The proposed method focuses on generating a smaller quantity of proposals while maintaining high recall, thereby enhancing the overall efficiency and efficacy of the detection process in a WSOD context. We apply the proposed WeakSAM on previous WSOD methods, including OICR [65] and MIST [53], which receive significant improvements. As shown in Table 1, quantitative results show that WeakSAM-enhanced WSOD can annotate bounding boxes for objects more precisely.

*Adaptive PGT Generation.* Generating high-quality pseudo ground truth (PGT) is the key to the WSOD paradigm. Traditional WSOD methods often encounter the issue of PGT incompleteness. This occurs because these methods typically select top-scoring proposals as PGT or apply a uniform threshold to filter proposals across all categories. Such approaches can lead to the omission of objects or entire categories, especially when proposals in certain categories score low. To address these problems, we propose an adaptive PGT generation method to normalize the score distribution of proposals, ensuring they fall within a similar range, as shown in Algorithm. 2.

For box list $B \in \mathbb{R}^{N \times 5}$ and corresponding score list $S \in \mathbb{R}^{N \times 1}$, we first select them with a specific classification label and then normalize the scores. The $N$ is the number of predicted boxes, and the second dimension of $B$ is the combination of a category label and four coordinate values. Next, we keep boxes with scores higher than the threshold $\tau_s$. Please note that the normalization enables the threshold to work for all categories adaptively, so we would not lose a ground truth category even if all boxes in this category have low scores. Then, we select the boxes whose main parts are not contained in some bigger boxes. Because the boxes that have more *overlap* are often local components of some objects. Last, we return the box list $B'$ as the final PGT.

**Algorithm 2** Adaptive Pseudo Ground Truth Generation

**Require:** boxes list $B$ of an image, corresponding scores list $S$, corresponding classification labels $Y$, score threshold $\tau_s$, overlap threshold $\tau_o$
**Ensure:** pseudo ground truth boxes $B'$
  1: initialize $B'$ as empty list
  2: **for** each $y_i$ in $Y$ **do**
  3:    // get boxes' indices with label $y_i$
  4:    $idx_i = \text{where} (B[:, 0] == y_i)$
  5:    $S_i = S[idx_i, :]$
  6:    $B_i = B[idx_i, :]$
  7:    $S_i^{norm} = \frac{S_i - \min(S_i)}{\max(S_i) - \min(S_i)}$ // normalize scores
  8:    // keep boxes with high score
  9:    $idx_{keep} = \{j \mid s_j \in S_i^{norm}, s_j > \tau_s\}$
10:    $B_i = B_i[idx_{keep}, :]$
11:    $S_i^{norm} = S_i^{norm}[idx_{keep}, :]$
12:    // select boxes with less overlap
13:    **for** each box $b_j$ in $B_i$ **do**
14:       $overlaps = \{ \frac{|b_j \cap b_k|}{|b_j|} \mid b_k \in B_i, k \ne j \}$
15:       **if** all $overlap < \tau_o$ in $overlaps$ **then**
16:          Append $b_j$ to $B'$
17:       **end if**
18:    **end for**
19: **end for**
20: **return** $B'$

*RoI Drop Regularization.* A recognized issue in the retraining phase of WSOD is noisy PGT instances. These noisy instances result in PGT acting as the inaccurate supervision. Alleviating this problem is critical for enhancing the performance of WSOD retraining. To analyze this problem in depth, we first divide the RoIs into different loss intervals. Then, we mark the RoIs whose corresponding PGTs do not have at least 70% IoU with the ground truth boxes as error ones. Last, we present the statistics as shown in Fig. 3, which demonstrates that the RoIs with larger losses are in a small amount and have a high error rate.

Intuitively, we propose a method, named RoI drop regularization, to adaptively drop the RoIs with larger losses. Notably, the proposed method is easy to implement and can further help the query-based detectors to alleviate the noisy PGT problem by its variant, query drop regularization. For anchor-based FSOD methods, e.g., Faster-RCNN [52], we first determine the thresholds $\tau_{cls}$ and $\tau_{reg}$ for classification loss and regression loss, respectively. Then, we compute the drop signal $d_i$ for $i$−th RoI.

$$d_i = \begin{cases} 1, & l_i^{cls} \le \tau_{cls}, \text{ and } l_i^{reg} \le \tau_{reg} \\ 0, & \text{others} \end{cases}, \quad (1)$$

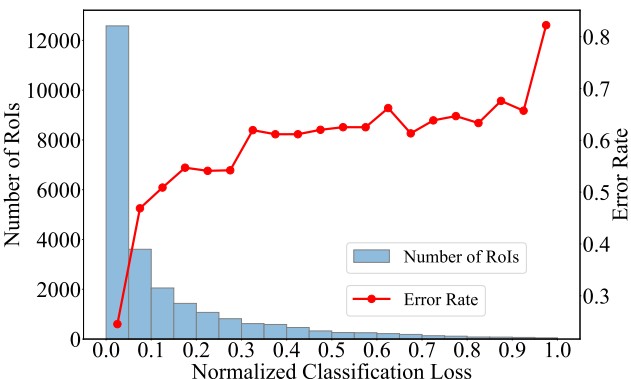

**Figure 3: The relationship between the normalized classification loss, corresponding number of RoIs and error rate. The results are obtained from training the Faster-RCNN using PGT in the preliminary training stage.**

where the $l_i^{cls}$ and $l_i^{reg}$ represent the classification loss and regression loss for each RoI, respectively. When the two losses of a RoI are all below their thresholds, we set its drop signal $d_i$ as 1. Finally, we integrate the $d_i$ into the computation of final loss $\mathcal{L}$.

$$\mathcal{L} = \sum_i d_i l_i^{cls} + \lambda \sum_i p_i^* d_i l_i^{reg}, \quad (2)$$

where $p_i^*$ is 1 if the box is positive, and 0 if the box is negative. The $\lambda$ is a balancing weight.

For query-based FSOD methods, e.g., DINO [82], since queries can be regarded as dynamic RoIs, we apply query drop regularization on them. Because only a few matched queries need to calculate box loss $l^{box}$ and IoU loss $l^{iou}$, we only set a percentile threshold based on classification loss $l^{cls}$. Only when the $i$−th query's loss $l_i^{cls}$ is less than the loss at $\tau\%$ percentile, i.e., $l_\tau^{cls}$, will its corresponding $d_i$ be set to 1.

$$d_i = \begin{cases} 1, & l_i^{cls} \leq l_\tau^{cls} \\ 0, & \text{others} \end{cases}. \quad (3)$$

$$\mathcal{L}_{\text{Hungarian}} = \sum_i d_i [l_i^{cls} + p_i^* l_i^{box} + p_i^* l_i^{iou}]. \quad (4)$$

## 3.3 WeakSAM for WSIS

Thanks to the high-quality WeakSAM PGT, we can directly use them to prompt SAM for precise segments as pseudo instance labels. Following the practices in the WeakSAM WSOD pipeline, we evaluate the quality of WeakSAM PGT using R-CNN-based and query-based instance segmentation methods, respectively. Notably, we do not introduce more techniques in the WeakSAM WSIS, because the WeakSAM pseudo instance labels are accurate enough.

## 4 EXPERIMENT

### 4.1 Experimental Setup

*Datasets and Metrics.* We evaluate the proposed WeakSAM on both weakly-supervised object detection (WSOD) and weakly-supervised

instance segmentation (WSIS) benchmarks. Notably, the same datasets for different tasks may have different settings.

**For WSOD**, we use three datasets, i.e., PASCAL VOC 2007 [14], PASCAL VOC 2012 [14], and COCO 2014 [44]. PASCAL VOC 2007 has 2501 images for training, 2510 images for evaluation, and 4592 images for testing. PASCAL VOC 2012 contains 5717 training images, 5823 validation images, and 10991 test images. COCO 2014 includes around 80,000 images for training and 40,000 images for validation. Following previous WSOD methods, we train WeakSAM on *train* and *val* sets and evaluate WeakSAM on the *test* set for PASCAL VOC 2007 and 2012. For COCO 2014, we use the *train* set for training and the *val* set for evaluating. PASCAL VOC 2007 and 2012 datasets comprise 20 object categories and COCO 2014 comprises 80 ones. We report the average precision AP metrics for these benchmarks.

**For WSIS**, we use two datasets, i.e., PASCAL VOC 2012, and COCO 2017. The PASCAL VOC 2012 dataset includes 10582 images for training, and 1449 images for evaluation, comprising 20 object categories. The COCO 2017 dataset includes 115K training images, 5K validation images, and 20K testing images, comprising 80 object categories. Following previous methods, we report the average precision AP metrics with different Intersection-over-Union (IoU) thresholds.

*Implementation Details.* For WeakSAM proposals generation, we adopt the WeakTr [92] with DeiT-S [68] model for generating classification clues, the SAM [34] with ViT-H [13] model to generate proposals. For WeakSAM WSOD pipeline, we use the WSOD networks, i.e., OICR [65], and MIST [53], with the VGG-16 [20] backbone to generate pseudo ground truth (PGT), and FSOD networks, i.e., Faster R-CNN [52] and DINO [82], with the ResNet-50 [22] backbone to retrain. As for the WeakSAM WSIS, we use SAM-ViT-H to generate pseudo instance labels and train the R-CNN-based and query-based methods, i.e., Mask R-CNN [21] and Mask2former [10], respectively. All hyper-parameters in Alg. 1 and Alg. 2 are following the default manners as Zhu et al. [92] and Sui et al. [60].

### 4.2 Comparisons with State-of-the-art Methods

*Weakly-supervised object detection.* We present the quantitative WSOD results in Table. 1. Compared with our WSOD baseline methods, i.e., OICR and MIST, the proposed WeakSAM achieves over 10% improvements on all metrics. The results of WeakSAM (MIST) surpass all WSOD methods on all metrics, which demonstrate the effectiveness of WeakSAM proposals. Compared with WSOD methods retrained by pseudo ground truth (PGT), the WeakSAM (MIST) with Faster R-CNN retraining still outperforms the SoS-WSOD [60] and W2N [26] on all metrics, and the WeakSAM (MIST) with DINO retraining even has comparable performance with fully-supervised Faster R-CNN. The retraining results demonstrate the effectiveness of the proposed WSOD pipeline, which includes the adaptive PGT generation and RoI drop retraining. Compared with concurrent work, WSOVOD [43], which also incorporates SAM, our WeakSAM (MIST) also achieves better performance.

*Weakly-supervised instance segmentation.* We first present the quantitative WSIS results of the PASCAL VOC 2012 *val* set in Table 2. The proposed WeakSAM with Mask R-CNN retraining achieves the best performance, which demonstrates the WeakSAM can benefit

**Table 1: Comparisons of the WSOD performance in terms of AP metrics on three benchmarks: PASCAL VOC 2007, PASCAL VOC 2012, and COCO 2014. The $Sup.$ column denotes the type of supervision used for training including full supervision ($\mathcal{F}$), point-level labels ($\mathcal{P}$), image-level labels ($\mathcal{I}$). "*" means the results rely on MCG [50] proposals. "‡" means this method use the a heavy RN50-WS-MRRP [58] backbone (1.76 × parameters than VGG16 and 10.10 × parameters than RN50). We mark the best WSOD results in bold.**

| Methods | Proposal | $Sup.$ | Retrain | VOC 07 AP$_{50}$ | VOC 12 AP$_{50}$ | COCO 14 AP$_{50:95}$ | AP$_{50}$ | AP$_{75}$ |
|---------|----------|--------|---------|------|------|------|------|------|
| *Fully-supervised object detection methods.* | | | | | | | | |
| Faster R-CNN [52] | RPN | $\mathcal{F}$ | – | 69.9 | – | 21.2 | 41.5 | – |
| *WSOD methods with point supervision.* | | | | | | | | |
| P2BNet [6] | RPN | $\mathcal{P}$ | – | 60.2 | – | 19.4 | 43.5 | – |
| *WSOD methods with image-level supervision.* | | | | | | | | |
| C-MIDN [18] | SS, MCG | | – | 52.6 | 50.2 | 9.6* | 21.4* | – |
| WSOD$^2$ [81] | SS | | – | 53.6 | 47.2 | 10.8 | 22.7 | – |
| SLV [9] | SS | | – | 53.5 | 49.2 | – | – | – |
| CASD [27] | SS | | – | 56.8 | 53.6 | 12.8 | 26.4 | – |
| IM-CFB [79] | SS | $\mathcal{I}$ | – | 54.3 | 49.4 | – | – | – |
| OD-WSCL [55] | SS, MCG | | – | 56.4 | 54.6 | 13.7* | 27.7* | 11.9* |
| WSOD-CBL [80] | SS | | – | 57.4 | 53.5 | 13.6 | 27.6 | – |
| WSOVOD [43] | LO-WSRPN + SAM | | – | 59.1 | 59.8 | 18.8 | 27.1 | 19.7 |
| WSOVOD‡ | LO-WSRPN + SAM | | – | 63.4 | 62.1 | 20.5 | 29.1 | 21.4 |
| *Baseline and ours.* | | | | | | | | |
| OICR [65] | SS, MCG | $\mathcal{I}$ | – | 41.2 | 37.9 | 8.0* | 18.9* | 7.0* |
| WeakSAM (OICR) | WeakSAM | | – | 58.9 +17.7 | 58.4 +20.5 | 19.9 +11.9 | 32.1 +13.2 | 20.6 +13.6 |
| *Baseline and ours.* | | | | | | | | |
| MIST [53] | SS, MCG | $\mathcal{I}$ | – | 54.9 | 52.1 | 11.4* | 24.3* | 9.4* |
| WeakSAM (MIST) | WeakSAM | | – | 67.4 +12.5 | 66.9 +14.8 | 22.9 +11.5 | 35.2 +10.9 | 24.6 +15.2 |
| *WSOD methods with image-level supervision. + Retrain* | | | | | | | | |
| W2F [88] | RPN | | Faster R-CNN | 52.4 | 47.8 | – | – | – |
| SoS-WSOD [60] | RPN | $\mathcal{I}$ | Faster R-CNN | 64.4 | 61.9 | 16.6 | 32.8 | 15.2 |
| W2N [26] | RPN | | Faster R-CNN | 65.4 | 60.8 | 15.9 | 33.3 | 13.4 |
| *Ours. + Retrain* | | | | | | | | |
| WeakSAM (OICR) | RPN | | Faster R-CNN | 65.7 | 62.9 | 22.3 | 36.5 | 23.0 |
| WeakSAM (MIST) | RPN | $\mathcal{I}$ | Faster R-CNN | 71.8 | 69.2 | 23.8 | 38.5 | 25.1 |
| WeakSAM (OICR) | – | | DINO | 66.1 | 63.7 | 24.9 | 36.9 | 26.8 |
| WeakSAM (MIST) | – | | DINO | **73.4** | **70.2** | **26.6** | **39.3** | **29.0** |

**Table 2: Comparisons of the WSIS performance in terms of AP metrics on PASCAL VOC 2012. The $Sup.$ column denotes the type of supervision used for training including mask supervision ($\mathcal{M}$), saliency maps ($\mathcal{S}$), image-level labels ($\mathcal{I}$), and SAM models ($\mathcal{A}$). We mark the best WSIS results in bold.**

| Methods | Backbone | $Sup.$ | Retrain | VOC 12 AP$_{25}$ | AP$_{50}$ | AP$_{70}$ | AP$_{75}$ |
|---------|----------|--------|---------|------|------|------|------|
| *Fully-supervised instance segmentation methods.* | | | | | | | |
| Mask R-CNN [21] | ResNet-101 | $\mathcal{M}$ | – | 76.7 | 67.9 | 52.5 | 44.9 |
| *WSIS methods with image-level supervision. + Retrain* | | | | | | | |
| WISE [36] | ResNet-50 | $\mathcal{I}$ | Mask R-CNN | 49.2 | 41.7 | – | 23.7 |
| IRNet [1] | ResNet-50 | $\mathcal{I}$ | Mask R-CNN | – | 46.7 | 23.5 | – |
| LIID [46] | ResNet-50 | $\mathcal{I} + \mathcal{S}$ | Mask R-CNN | – | 48.4 | – | 24.9 |
| Arun et al.[3] | ResNet-50 | $\mathcal{I}$ | Mask R-CNN | 59.7 | 50.9 | 30.2 | 28.5 |
| WS-RCNN [49] | VGG-16 | $\mathcal{I}$ | Mask R-CNN | 62.2 | 47.3 | – | 19.8 |
| BESTIE [33] | HRNet-W48 | $\mathcal{I}$ | Mask R-CNN | 61.2 | 51.0 | 31.9 | 26.6 |
| CIM [41] | ResNet-50 | $\mathcal{I}$ | Mask R-CNN | 68.7 | 55.9 | 37.1 | 30.9 |
| *Ours.* | | | | | | | |
| WeakSAM | ResNet-50 | $\mathcal{I} + \mathcal{A}$ | Mask R-CNN | 70.3 | 59.6 | 43.1 | 36.2 |
| WeakSAM | ResNet-50 | $\mathcal{I} + \mathcal{A}$ | Mask2Former | **73.4** | **64.4** | **49.7** | **45.3** |

WSIS effectively. Furthermore, the pseudo instance labels generated by WeakSAM can also be used by the modern query-based methods, e.g., Mask2Former [10], which achieves the best results.

We then show the quantitative WSIS results on COCO 2017 *val* and *test* sets. On these more challenging benchmarks, WeakSAM with Mask R-CNN retraining achieves better results than CIM [41].

**Table 3: Comparisons of the WSIS performance in terms of AP metrics on COCO 2017. The *Sup.* column denotes the type of supervision used for training including mask supervision ($\mathcal{M}$), saliency maps ($\mathcal{S}$), image-level labels ($\mathcal{I}$), and SAM models ($\mathcal{A}$). We mark the best WSIS results in bold.**

| Methods | Backbone | *Sup.* | Retrain | COCO val 2017 | | | COCO test-dev | | |
|---|---|---|---|---|---|---|---|---|---|
| | | | | AP$_{50:95}$ | AP$_{50}$ | AP$_{75}$ | AP$_{50:95}$ | AP$_{50}$ | AP$_{75}$ |
| *Fully-supervised instance segmentation methods.* | | | | | | | | | |
| Mask R-CNN [21] | ResNet-50 | $\mathcal{M}$ | – | 34.4 | 55.1 | 36.7 | 33.6 | 55.2 | 35.3 |
| *WSIS methods with image-level supervision.* | | | | | | | | | |
| WS-JDS [59] | VGG-16 | $\mathcal{I}$ | – | 6.1 | 11.7 | 5.5 | – | – | – |
| PDSL [57] | ResNet18-WS | $\mathcal{I}$ | – | 6.3 | 13.1 | 5.0 | – | – | – |
| Fan et al. [15] | ResNet-101 | $\mathcal{I} + \mathcal{S}$ | Mask R-CNN | – | – | – | 13.7 | 25.5 | 13.5 |
| LIID [46] | ResNet-50 | $\mathcal{I} + \mathcal{S}$ | Mask R-CNN | – | – | – | 16.0 | 27.1 | 16.5 |
| BESTIE [33] | HRNet-W48 | $\mathcal{I}$ | Mask R-CNN | 14.3 | 28.0 | 13.2 | 14.4 | 28.0 | 13.5 |
| CIM [41] | ResNet-50 | $\mathcal{I}$ | Mask R-CNN | 17.0 | 29.4 | 17.0 | 17.2 | 29.7 | 17.3 |
| *Ours.* | | | | | | | | | |
| WeakSAM | ResNet-50 | $\mathcal{I} + \mathcal{A}$ | Mask R-CNN | 20.6 | 33.9 | 22.0 | 21.0 | 34.5 | 22.2 |
| WeakSAM | ResNet-50 | $\mathcal{I} + \mathcal{A}$ | Mask2Former | **25.2** | **38.4** | **27.0** | **25.9** | **39.9** | **27.9** |

**Table 4: Ablation studies for WeakSAM prompts on PASCAL VOC 2007. We evaluate the average number of proposals, recall, and WSOD performance by MIST [53].**

| SS | Dense Sample | CAM$_{fine}$ | CAM$_{coarse}$ | Cross Attn. | Num. | Recall | | | AP$_{50}$ |
|---|---|---|---|---|---|---|---|---|---|
| | | | | | | IoU=0.50 | IoU=0.75 | IoU=0.90 | |
| ✓ | | | | | 2001 | 92.6 | 57.7 | 19.2 | 54.9 |
| | ✓ | | | | 129 | 79.6 | 50.7 | 24.3 | 45.2 |
| | ✓ | ✓ | | | 151 | 88.9 | 67.0 | 37.2 | 63.3$_{+18.1}$ |
| | ✓ | ✓ | ✓ | | 174 | 90.6 | 70.1 | 40.1 | 65.5$_{+20.3}$ |
| | ✓ | ✓ | ✓ | ✓ | 213 | 95.6 | 75.0 | 42.1 | 67.4$_{+22.2}$ |

Besides, the WeakSAM with Mask2Former also presents the best results.

**Table 5: Ablation studies for adaptive PGT generation and RoI drop regularization. We present the results on the PASCAL VOC 2007 *test* set.**

**(a) Ablation studies for the anchor-based detector, i.e., Faster R-CNN [52].**

| Top-1 PGT | Adaptive PGT | RoI Drop | AP$_{50}$ |
|---|---|---|---|
| ✓ | | | 68.4 |
| | ✓ | | 70.7$_{+2.3}$ |
| | ✓ | ✓ | 71.8$_{+3.4}$ |

**(b) Ablation studies for the query-based detector, i.e., DINO [82].**

| Top-1 PGT | Adaptive PGT | Query Drop | AP$_{50}$ |
|---|---|---|---|
| ✓ | | | 71.1 |
| | ✓ | | 72.8$_{+1.7}$ |
| | ✓ | ✓ | 73.4$_{+2.3}$ |

## 4.3 Ablation Studies

In this section, we present the ablation studies to evaluate the improvements brought by the proposed methods, i.e., WeakSAM prompts, adaptive PGT generation, and RoI drop retraining.

Due to the limitation of pages, we leave more ablation studies in the supplementary material, including additional efficiency analysis, sensitivity analysis, qualitative analysis, discussions, etc.

**Table 6: Efficiency comparison between Selective Search and our WeakSAM during the training on the PASCAL VOC 2007. 'Num.' is the number of proposals, 'T$_{Proposals}$' is the time consumption for generating proposals, 'T$_{WSOD}$' is the time consumption for training the WSOD network, i.e., MIST [53], and 'M$_{WSOD}$' is the GPU memory cost for each GPU card.**

| | Num. | T$_{Proposals}$ | T$_{WSOD}$ | M$_{WSOD}$ |
|---|---|---|---|---|
| SS [69] | 2001 | 11.6 hrs | 16 hrs | 17810 MiB |
| Ours | 213$_{-89.4\%}$ | 4 hrs$_{-65.5\%}$ | 9 hrs$_{-43.8\%}$ | 5667 MiB$_{-68.2\%}$ |

*Improvements of WeakSAM Prompts.* To further analyze the improvements brought by the proposed WeakSAM prompts, we conduct ablation experiments for different prompts as shown in Table 4. Here, we use the Selective Search [69] as the baseline method and list the proposals' number, recall, and corresponding WSOD performance. When only using the densely sampled points as SAM prompts, the generated proposals can achieve 5.1% higher Recall (IoU=0.90), and 9.7% lower AP$_{50}$ for MIST. After adding peak CAM points and peak cross attention points as prompts, we can achieve

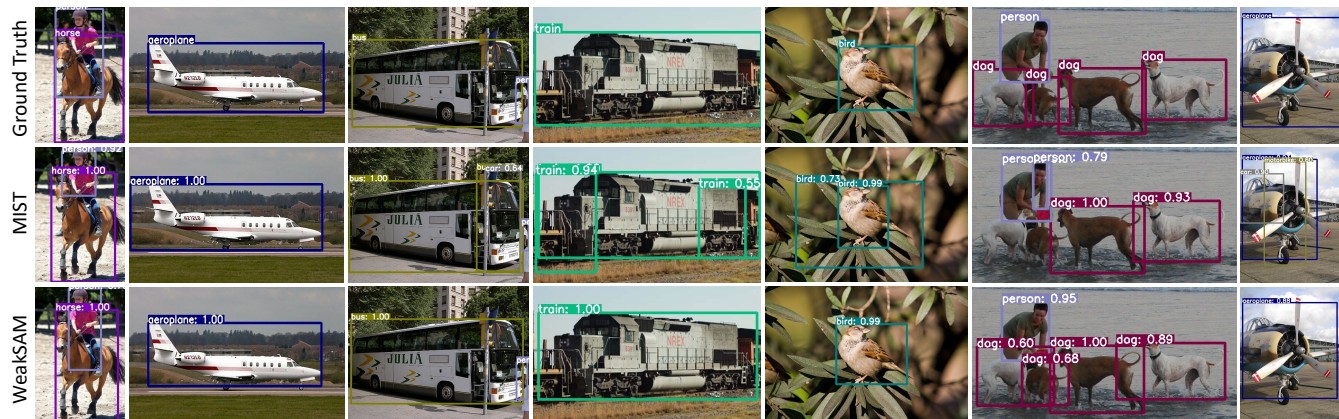

**Figure 4: Visualization of the weakly-supervised object detection on the PASCAL VOC 2007 *test* set.**

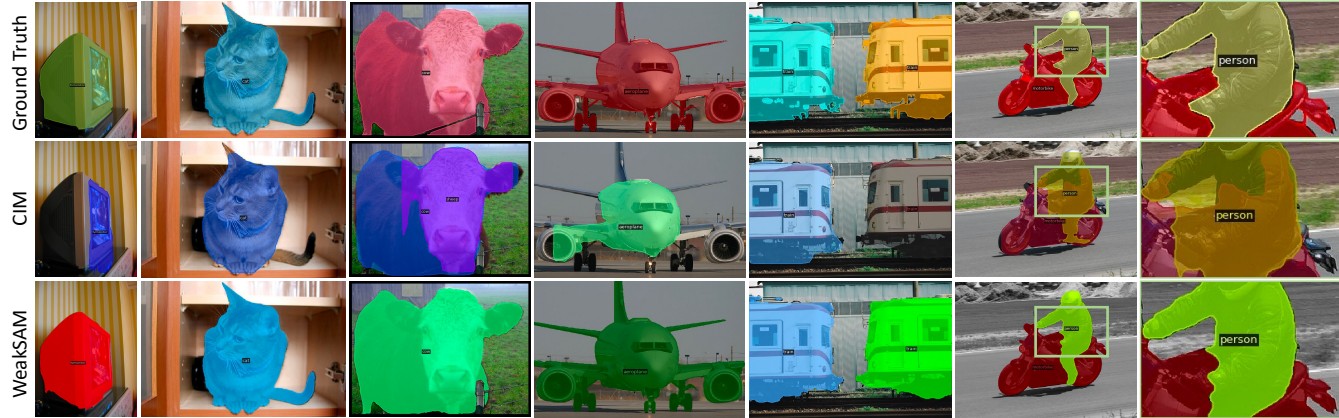

**Figure 5: Visualization of the weakly-supervised instance segmentation on the PASCAL VOC 2012 *val* set.**

higher recall and $AP_{50}$ through only 213 proposals on average. The results demonstrate the effectiveness of WeakSAM prompts.

*Improvements of WSOD Pipeline.* To further analyze the improvements brought by the proposed WeakSAM WSOD pipeline, we conduct ablation experiments for adaptive PGT generation and RoI drop regularization in Table 5. Here, we follow the common practice to set a baseline that uses the predicted boxes with the top-1 score as PGT and plain Faster R-CNN as the retraining network. The results show that both adaptive PGT generation and RoI drop regularization can help improve the $AP_{50}$ of the detector. Furthermore, both the RoI-based detector, Faster R-CNN [52] and query-based detecotr, DINO [82], can benefit from the proposed WSOD techniques.

### 4.4 Efficiency Comparison

To further analyze the efficiency improvement brought by our WeakSAM, we present the efficiency comparison between Selective Search [69] and our WeakSAM on a machine with 4 GPU cards, as shown in Table 6. Our WeakSAM reduces the number of proposals by 89.4%, the proposal generation time by 65.5%, the WSOD network training time by 43.8%, and the GPU memory cost by 68.2%. The results demonstrate the significant efficiency improvement brought by the proposed WeakSAM.

### 4.5 Visualization Results

Fig.4 presents the object detection results using WeakSAM (MIST), showing its capability to accurately capture entire objects without generating excessive noisy bounding boxes. In Fig.5, the instance segmentation results of WeakSAM Mask2Former retraining are showcased. The results indicate effective segmentation of entire instances with a notable reduction in overlapping segments.

## 5 CONCLUSION

In this paper, we introduce WeakSAM, a novel framework utilizing the Segment Anything Model (SAM) for weakly-supervised instance-level recognition, demonstrating leading performance in WSOD and WSIS benchmarks. Different from the original SAM, which requires interaction and can not be aware of categories, WeakSAM represents an innovative fusion of SAM with weakly-supervised learning (WSL), overcoming the redundancy problem of WSOD proposals. To further address WSOD issues such as pseudo ground truth (PGT) incompleteness and noisy PGT instances, our approach includes adaptive PGT generation and a Region of Interest (RoI) drop regularization. The adaptability of WeakSAM is further showcased through its extension to weakly-supervised instance segmentation (WSIS). Our work aims to inspire further research with SAM and WSL, contributing significantly to the development of a universal framework for weakly-supervised recognition.

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
