# OpenReview forum: "WeakSAM: Segment Anything Meets Weakly-supervised Instance-level Recognition"
_acmmm.org/ACMMM/2024/Conference — MM2024 Poster_

### Official Review · Reviewer_ASBq · 2024-05-24

**Rating:** 4
**Confidence:** 2

**Summary:**

This paper proposed WeakSAM, it solves the weakly-supervised object detection (WSOD) and segmentation by utilizing the pre-learned world knowledge contained in a vision foundation model. It addresses the SAM's shortcomings of requiring human prompts and category unawareness in object detection and segmentation. Quantitive experiments demonstrate the effectiveness of the model.

**Strengths:**

1. The proposed method can exceed previous methods by a large margin.
2. The paper is well-written and is easy to read.

**Limitations:**

SAM is well-trained using a lot of labeled/unlabeled data.  I am wondering if it is fair to compare the previous method, using such a powerful auxiliary model to generate pseudo-labels. What about the performance of the other models if using the SAM model. If not using the powerful SAM model, can the model perform well?

**Suitability:**

3

---

### Official Review · Reviewer_YbAk · 2024-05-24

**Rating:** 5
**Confidence:** 3

**Summary:**

This paper proposes WeakSAM and solves the weakly-supervised object detection (WSOD) and segmentation by utilizing the pre-learned world knowledge contained in a vision foundation model, i.e., the Segment Anything Model. The proposed method can address the issues of pseudo ground truth (PGT) incompleteness and the presence of noisy PGT instances by introducing adaptive PGT generation and region of Interest (RoI) drop regularization, respectively.

**Strengths:**

1. The paper is well-written and is easy to follow.
2. The motivation of this paper is that assimilating SAM within the WSL paradigm is interesting.
3. The SOTA experimental results show the effectiveness of the proposed WeakSAM.

**Limitations:**

It is recommended to try the WSOD model in recent years as a baseline.

**Suitability:**

3

---

### Official Review · Reviewer_DBpV · 2024-06-02

**Rating:** 2
**Confidence:** 4

**Summary:**

This paper introduces the SAM model into weakly supervised object detection and designs two modules: adaptive PGT generation and RoI drop regularization. I think the major novelty of this paper appears to be a unified innovation, i.e., putting together many existing techniques into a unified framework. The novelty of the two newly proposed modules is limited.

**Strengths:**

Strong results.

**Limitations:**

1. The proposed framework seems to be a unified innovation. The novelty of adaptive PGT generation and RoI drop regularization modules is limited.
2. In adaptive PGT generation, the authors normalize candidate proposals of each category to enhance system performance. I wonder whether this strategy is also effective in previous WSOD frameworks, i.e., without using the proposals generated by SAM. I am not entirely convinced of the effectiveness of this module, and I consider that the major efficacy is attributed to the object localization capability of SAM.
3. More visualization results should be provided without using the two proposed modules, which can demonstrate the effectiveness of the designing modules.

**Suitability:**

3

---

### Official Review · Reviewer_Urx4 · 2024-06-07

**Rating:** 3
**Confidence:** 3

**Summary:**

This paper introduces WeakSAM and leverages the pre-learned world knowledge contained in a vision foundation model SAM to solve the weakly supervised object detection (WSOD) and segmentation problems.

**Strengths:**

The WeakSAM framework proposed in this paper integrates the Segment Anything Model (SAM) into weakly supervised learning (WSL), improving the performance of weakly supervised object detection (WSOD) and instance segmentation (WSIS).

The experimental results show that WeakSAM performs significantly better than previous state-of-the-art methods in multiple benchmarks, demonstrating its effectiveness.

**Limitations:**

(1) Regarding "weak supervision": Let's briefly forget about SAM. If a class-agnostic proposal generator (e.g., RPN) is trained using a large amount of fully labeled data and then applied to the WSOD pipeline, can this still be considered weak supervision?

(2) It seems necessary to separate detection into classification and localization. The classification task can be viewed as weakly supervised, while localization benefits significantly from SAM, which is fully supervised. Although the paper title indicates weakly supervised recognition, this seems blurred in the experimental section. In other words, the paper should clearly separate classification and localization and demonstrate how much gain the fully supervised localization capability brings.

(3) The paper conducts many comparisons with SS, which may not be very necessary. As mentioned in (2), the gains are primarily due to SAM. Therefore, the paper should focus on how to effectively utilize SAM, for example, by exploring various ways to trigger SAM to generate higher-quality proposals.

**Suitability:**

2

---

### Meta-Review · Area_Chair_yZ7V · 2024-06-28

**Recommendation:** Accept (Poster)
**Confidence:** 5

**Metareview:**

Initially, the paper received one Weak Reject, one Borderline Reject, one borderline Accept, and one Weak Accept. The reviewers praised the proposed approach's effectiveness but raised significant concerns regarding the claim of weak supervision and the fairness of comparisons.

The rebuttal addressed these points by providing additional experimental results and clarifications. After the rebuttal, Reviewer DBpV upgraded the score from Weak Reject to Borderline Accept, confirming the method's novelty of combining vision foundation models for WSOD. Reviewer YbAk maintained Weak Accept for the good performance and interesting motivation. Reviewer Urx4 decreased the score and expressed concerns about the fairness and reasonability of the WSOD setting.

The AC has read the paper, reviews, and rebuttal. The AC agrees with Reviewer Urx4 that exploiting large-scale pretrained models like SAM can be unfair in comparison with traditional methods. However, the AC believes that this limitation may not overturn the overall rating of this paper, considering its good performance and novel pipeline for addressing WSOD with foundation models.

The authors should carefully improve the final paper by following reviewer recommendations, particularly as suggested by Reviewer Urx4.